# A model of antibiotic resistance genes accumulation through lifetime exposure from food intake and antibiotic treatment

Henry Todman[1], Sankalp Arya[1¤a], Michelle Baker[1¤b], Dov Joseph Stekel[1,2]*

**1** School of Biosciences, University of Nottingham, Sutton Bonington Campus, Loughborough, Nottingham, United Kingdom, **2** Department of Mathematics and Applied Mathematics, University of Johannesburg, Rossmore, South Africa

¤a Current address: Department of Population Health and Pathobiology, North Carolina State University, Raleigh, NC, United States of America
¤b Current address: School of Veterinary Medicine and Science, University of Nottingham, Sutton Bonington Campus, Loughborough, Nottingham, United Kingdom
* dov.stekel@nottingham.ac.uk

## Abstract

Antimicrobial resistant bacterial infections represent one of the most serious contemporary global healthcare crises. Acquisition and spread of resistant infections can occur through community, hospitals, food, water or endogenous bacteria. Global efforts to reduce resistance have typically focussed on antibiotic use, hygiene and sanitation and drug discovery. However, resistance in endogenous infections, e.g. many urinary tract infections, can result from life-long acquisition and persistence of resistance genes in commensal microbial flora of individual patients, which is not normally considered. Here, using individual based Monte Carlo models calibrated using antibiotic use data and human gut resistomes, we show that the long-term increase in resistance in human gut microbiomes can be substantially lowered by reducing exposure to resistance genes found food and water, alongside reduced medical antibiotic use. Reduced dietary exposure is especially important during patient antibiotic treatment because of increased selection for resistance gene retention; inappropriate use of antibiotics can be directly harmful to the patient being treated for the same reason. We conclude that a holistic approach to antimicrobial resistance that additionally incorporates food production and dietary considerations will be more effective in reducing resistant infections than a purely medical-based approach.

## Introduction

The human gut is a diverse and dynamic environment playing host to a wide variety of bacteria, viruses, archaea and eukaryotes. Genomic studies have shown that the human gut microbiota contains more than 300 species of bacteria [1]. These commensal enteric bacteria are thought to be largely harmless and play an important role in maintaining the health of the host through various mechanisms: i.e. protection against colonisation by pathogens through competitive exclusion or production of antimicrobial chemicals [2–4], maintaining the host

**Data Availability Statement:** This is a modelling paper. No primary data was generated. The model code used for simulations is included as Supporting information.

**Funding:** MB and DJS received funding from the Natura Environment Research Council, UK, under grant number NE/N019881/1. The funders played no role in the study design, data collection and analysis, decision to publish or preparation of the manuscript.

**Competing interests:** The authors have declared that no competing interests exist.

immune system [3, 5], and extracting energy and nutrients from food [3]). However, at lower levels there are pathogenic bacteria endogenous to the gut (in particular *Enterobacteriaceae* and *Enterococcaceae*) [6].

In addition to playing host to a diverse collection of bacteria and other micro-organism, the human gut represents a reservoir of antimicrobial resistance (AMR), with antimicrobial resistance gene (ARG) abundance known to be correlated at a population scale with antibiotic use [7]. Enteric bacteria hold ARGs either on their chromosomes or on mobile genetic elements; it is thought that these ARGs predominantly reside within non-pathogenic unclassified species [8].

Antimicrobial resistances in the gut may become established after ingestion of contaminated food. Many studies have shown ARGs to be present in a range of high-risk food products: raw and cooked meats [9–13], fermented milk products [14–16], fermented meat products [17, 18] and vegetable products [19–21]. The ready-to-eat food market is particularly problematic due to a lack of cooking and washing before consumption [22–25].

Several authors have identified systematic differences in ARG levels between individuals resident in different countries [7, 26, 27]. One source of these differences is likely to result from differing levels of availability of antibiotic treatments. Governmental approaches to antibiotic availability are diverse and defined daily doses per inhabitant can vary widely [7, 28, 29].

Ageing individuals are at higher risk of AMR infections because of increased exposure to ARGs, increased lifetime exposure to antibiotics and increased vulnerability to infection with age. The number of ARGs within an individual's intestinal tract is correlated with age [30, 31]. Lu *et al.* [30] showed that the number of resistances from faecal samples of four different age groups were positively correlated with increasing age. Further, cluster analysis suggested that resistances were being acquired and accumulated over time rather than being transient. Other research has also supported this view. Ghosh *et al.* [8] profiled resistance genes of 275 gut flora samples sourced from multiple countries and found increasing ARG diversity with age. Antimicrobial resistant bacterial infections in older age can often result from endogenous bacteria moving from the intestinal tract to other areas of the body, for example the urinary tract [32, 33].

In this work, we bring together these aspects of ARG establishment using a probabilistic mathematical model of the accumulation of antimicrobial resistance over an individual lifetime. We consider four key model parameters: antibiotic use, ARG ingestion, and the probabilities of an ARG becoming established in the presence and absence of concurrent antibiotic use. We use the model to identify which of these factors, individually and in combination, are most important in explaining the accumulation of ARGs during an individual's lifetime, and so identify potential mitigations against this aspect of antimicrobial resistance.

## Methods

We have defined a probabilistic model to define the acquisition of ARGs in the enteric system of individuals (Fig 1), which we have evaluated using Monte Carlo simulations. We consider the acquisition of resistance genes to 14 different classes of antibiotics. In order to reduce model complexity, we consider resistance to individual classes of antibiotics (e.g. penicillins, carbapenems, cepholosporins, aminoglycosides, etc), rather than specific antibiotics. This is a purposefully simplified view of resistance to different antibiotics: there can be many different genes conferring resistance to antibiotics in the same class, and even many different genes conferring resistance to the same antibiotic (e.g. there are over 40 different genes divided into 11 different classes of action conferring resistance to tetracycline [34]).

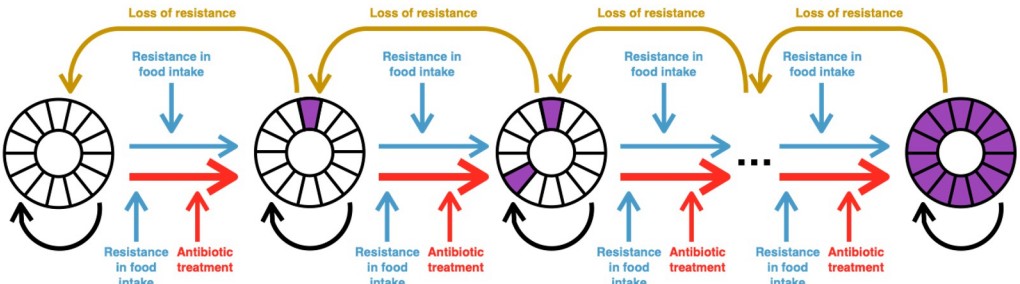

**Fig 1. Schematic diagram of the lifetime food model showing key model interactions.**

For each antibiotic class, we consider the probability that an individual is exposed to resistance genes through ingestion of food, and the probability that a resistance gene becomes fixed in the individual's enteric bacterial communities. The probability of resistance becoming established in the gut microbiome is dependant on whether the individual may be undergoing antibiotic treatment (of the same class as the resistance genes), as the presence of antibiotic treatment provides selective pressure for these resistance genes. For each antibiotic class, we also consider the probability of exposure to resistance genes for that antibiotic via the food chain, as well as the probability of resistance becoming established in the gut microbiome in the presence and absence of selective pressures from concurrent antibiotic treatment.

Throughout our analysis we considered three different levels of antibiotic use, which reflect different national levels of antibiotic availability. We ran different simulations of this model, with each scenario for 1000 individuals. We have based the parameter values for the probability of antibiotic use in areas with low and medium antibiotic usage from drug utilisation figures for European countries [26], and then estimated high antibiotic use areas parameter values based on this.

We simulate individual lifetimes in the Monte Carlo model with time steps of one week. The probability of an individual being exposed to resistance via food intake each week ($P_{\text{FoodRes.}}$) is a random variable with a uniform distribution. Once exposed to resistance, there is a probability that this resistance will establish in the microbial flora in the gut of the individual ($P_{\text{Fix}}$). Each week, there is an independent probability that the individual may undergo antibiotic treatment ($P_{\text{Ab. Treat.}}$). As the use of antibiotics can exert selective pressures on resistant bacterial populations, we assign a greater probability of establishment of ARGs in the presence of antibiotic treatment ($P_{\text{Ab. Fix.}}$). The probability that the individual acquires a new class of resistance in any given week is given by the transition probability (1). The majority of simulations were performed without antibiotic resistance loss; in order to simulate the possible loss of ARGs from the resistome through wash out, at the end of each time step in the Markov chain model there is a possibility that an acquired resistance, $A_i$, is lost with probability $P_{\text{Loss}}(A_i)$.

$$\mu_i = P_{\text{food res.}}(A_i)((1 - P_{\text{Ab. treat}}(A_i))P_{Fix}(A_i) + P_{\text{Ab. treat}}(A_i)P_{Ab.Fix}(A_i)) \qquad (1)$$

The parameters used for each scenario are given in the Table 1. At each time step in the model, we sample the probability of resistance in food intake from a continuous uniform distribution $U(0, 0.5)$: this distribution is chosen to reflect intake of a varied diet from a variety of food sources, with the upper bound based on observed frequencies of resistance in *Escherichia coli* isolates from food products [35, 36]. *E. coli* data were used here because this is a standard

**Table 1. The standard parameters used when simulating the lifetime resistance model.**

| Parameter | Parameter Name | Parameter Values | Parameter Range |
|---|---|---|---|
| $N$ | Number of antibiotic classes considered | 14 | - |
| $n$ | Number of individuals simulated | 1000 | - |
| $P_{\text{Ab. Treat}}(A_i)$ | Probability of an individual receiving antibiotic treatment for antibiotic class $A_i$ | $1.4 \times 10^{-2}$ (low Ab usage) $2.1 \times 10^{-2}$ (medium Ab usage) $5.0 \times 10^{-2}$ (high Ab usage) | 0—1 |
| $\beta_{\text{food}}$ | Upper bound for uniform distribution of $P_{\text{food res.}}(A_i)$ | 0.5 | 0—1 |
| $P_{\text{food res.}}(A_i)$ | Probability of resistance genes for antibiotic class $A_i$ being present in an individual's food intake | $P_{\text{food res.}}(A_i) \sim \mathcal{U}(0, \beta_{\text{food}})$ | $0—\beta_{\text{food}}$ |
| $P_{\text{Fix}}(A_i)$ | Probability of resistance genes becoming established in an individuals resistome in absence of antibiotics | $1.0 \times 10^{-4}$ | 0—1 |
| $P_{\text{Ab. Fix}}(A_i)$ | Probability of resistance genes for antibiotic class $A_i$ becoming established in an individuals resistome in presence of antibiotics | $5.0 \times 10^{-2}$ | 0—1 |
| $P_{\text{Loss}}(A_i)$ | Probability of resistance genes for antibiotic class $A_i$ being lost | $1.0 \times 10^{-6}$ | 0—1 |

sentinel organism used for food safety testing; however, our resistance model is intended to me more generic. Then we estimate probabilities of resistance becoming established for this model based on metagenomic data for human gut microbiota in countries with different levels of antibiotic use [30].

Three different forms of sensitivity analyses were conducted. First, For local sensitivity analysis for each of the model parameters, we took 1000 parameter values sampled from the feasible parameter space (Table 1) and calculate the mean resistance load for each. Second, more detailed histograms were produced for 20% and 50% reduction in key parameters; these percentages were chosen to represent moderate and large reductions at a scale where we would expect to see an effect. Third, we also tested sensitivity to gene loss, by constructing an alternative model which includes the possibility of acquired resistance genes to be lost. MATLAB R2020b was used to run time course simulations of the lifetime resistance model and to perform sensitivity analyses for model parameters. Matlab code for the model is provided as file S1 Data.

## Results & discussion

Simulation of 1000 individuals in each of the three antibiotic use scenarios (low, medium and high) for the standard parameter set (Table 1) shows that higher antibiotic use increases ARG acquisition over time by commensal bacteria (Fig 2).

The average resistance loads in individuals, at 70 years of age, from medium and high antibiotic use countries, are respectively 24.38% and 71.40% higher than in low antibiotic use countries. Thus our model concurs with many researchers and organisations including WHO [37], in advocating for a reduction in antibiotic use as a way to control the spread of antibiotic resistance, among other strategies. This strategy reduces opportunities for resistance genes to be the selected for in the gut over the lifetime of the individual.

In order to frame analyses of potential interventions on long term accumulation of resistance, we carried out local sensitivity analyses of the key parameters from the model (Fig 3). The accumulation of resistance is most sensitive to the probability of antibiotic treatment, and the probability of resistance fixing during antibiotic use. This effect is greatest in low and medium use countries. This led us to investigate impact of those factors in greater detail.

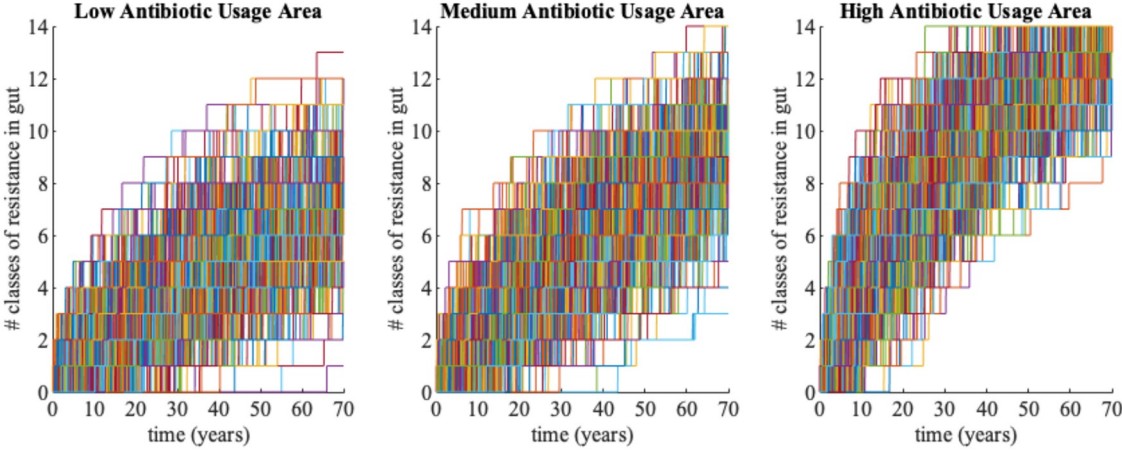

(a) Timecourse of lifetime resistance model

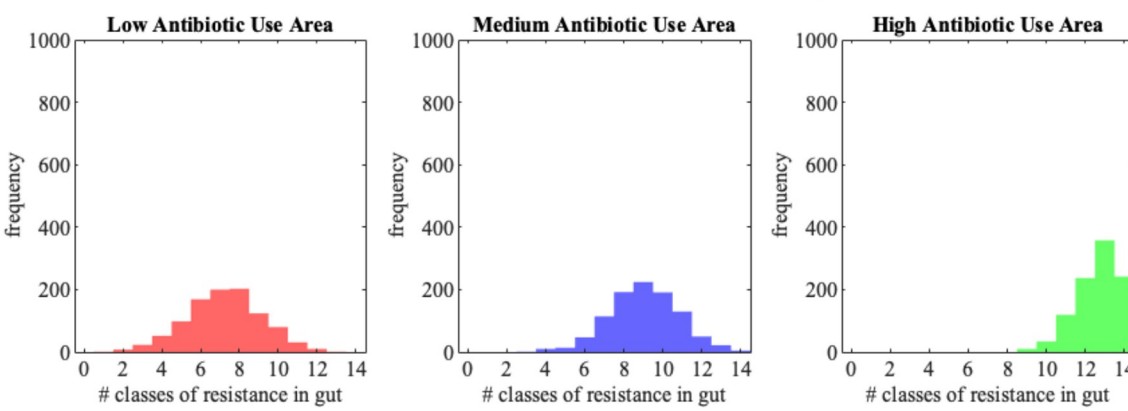

(b) Histogram showing the distribution of ARG load in individuals' resistomes

**Fig 2. (a) Time course simulation of the lifetime resistance model for low, medium and high antibiotic use countries**. In each of the three antibiotic use scenarios (low, medium and high), we have run the lifetime resistance model using the standard parameter set (given in Table 1) for 1000 individuals. Each line represents an individual simulated in the lifetime resistance model. We can clearly see that individuals acquire more ARGs more quickly in areas of higher antibiotic usage. **(b) Histogram showing the distribution of ARG load in individual's resistomes by age 70 for the lifetime resistance model**. These histograms show the distribution of the number of resistance classes at the end of the time course simulations of 1000 individuals shown in (a) (i.e. at age 70). The means and standard deviations, ($\mu$, $\sigma$), for low, medium and high antimicrobial use areas are (7.2120, 1.9475), (9.0410, 1.7946) and (12.6250, 1.1321) respectively.

### The number of resistance genes acquired by an individual is dependent on the use of antibiotics over the individual's lifetime and can be meaningfully reduced by a reduction in an individual's intake of ARGs through food

We further explored the impact of a reduction in Ab usage by running simulations with a 20% and 50% reduction in the probability of an individual undergoing treatment in any given week (Fig 4). This analysis explores the practical effects of an Ab reduction policy in countries such as Denmark (low usage), Spain (medium usage) or China (high usage). Reducing the antibiotic consumption in areas that have a higher rate of antibiotic treatment is on average more effective at reducing the ARG load: a 20% reduction in Ab usage

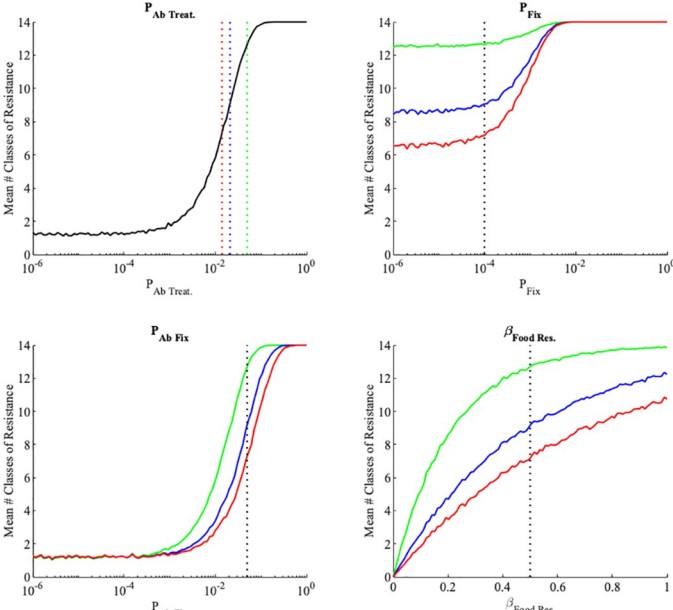

**Fig 3. Local sensitivity analysis of the lifetime resistance model parameters.** We vary the model parameters ($P_{Ab.\ Treat.}$, $P_{Fix}$, $P_{Ab.\ Fix}$, and $\beta_{food}$) across the possible parameter space (given in Table 1) and then calculated the mean ARG load at age 70 of 1000 individuals for each of the different parameter values. For $P_{Ab.\ Treat.}$, the black line shows the mean resistance load as the probability of undergoing antibiotic treatment is varied across the parameter space, and the dashed red, blue and green lines indicate the parameter values used for $P_{Ab.\ Treat.}$ in the low, medium and high antibiotic use areas respectively. For the local sensitivity analyses of $P_{Fix}$, $P_{Ab.\ Fix}$, and $\beta_{food\ res.}$, the red, blue and green lines represent the average resistance load as the parameter of interest is varied for low, medium and high antibiotic use areas respectively. The dashed black line in these subplots represents the values used for these parameters in the model simulations.

yields an average 5.98% and 11.33% reduction in resistance load by age 70 in high and medium use areas respectively, while a 50% reduces the mean resistance load by 21.21% and 32.92%.

An alternative avenue of control is reduction of ARG intake through food and water. As before, we considered two levels of reduction of ARGs in food, 20% and 50% for both antibiotic usage and ARG levels in food. All scenarios were applied for low, medium and high antibiotic use countries. Reduction of ARGs in overall food intake is especially effective in low and medium antibiotic use countries Fig 5, with 14% and 13% ARG load reduction with reduced Abs in food of 20% in medium and low use countries respectively, and 40% and 37% ARG load reduction for 50% food intake reduction. In high antibiotic use countries the impact is lower, with 6% and 23% reduction in ARG loads for 20% and 50% reduction in food ARGs respectively. Similar outcomes are observed when reducing ARGs in food only during antibiotic treatment, representing dietary change during such treatment, e.g. avoiding higher risk or raw foods. In that case, we saw a similar reduction in ARG load as for the reduced probability of ARG in food in general (Fig 6, S1 and S2 Figs).

The greatest reduction in the number of resistance classes acquired by age 70 comes from a combined approach. Here, for even a modest 20% decrease in both actions, we see a reduction of between 12.63% and 24.02%, depending upon the original level antibiotic usage. For a 50% decrease in both, we observe the number of resistance classes acquired by age 70 can be reduced significantly (by between 46.35% and 56.52%) reducing the likelihood that endogenous bacterial infections in older age will be resistant to treatment.

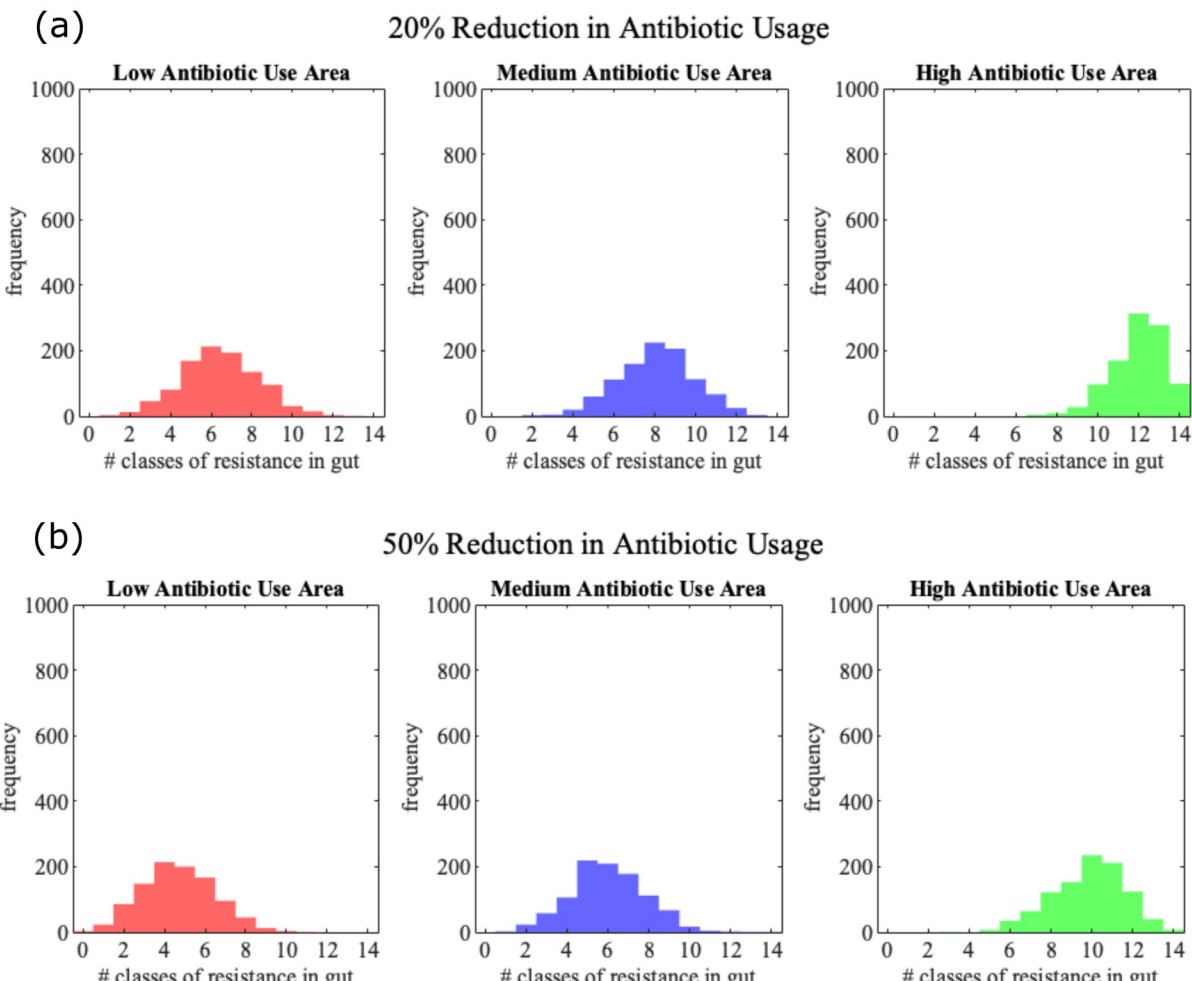

**Fig 4. Histograms showing distribution of ARG load in individual by age 70 when there is (a) 20% and (b) 50% reduction in the probability of antibiotic treatment.** The means and standard deviations, $(\mu, \sigma)$, for the low antibiotic use areas are (6.4400, 1.9042) and (4.6970, 1.7985) for 20% and 50% reduction respectively. Similarly $(\mu, \sigma)$ for the medium and high antibiotic use areas are (8.0660, 1.8604) and (11.9670, 1.3243) respectively when antibiotic use is reduced by 20%, and (6.0160, 1.8160) and (9.8370, 1.7770) respectively when reduced by 50%.

## Inclusion of ARG loss from resistome does not materially impact lifetime resistance model simulations for realistic values of $P_{\text{Loss}}$

We adapted the lifetime model to include the possibility of resistance loss due to ARG washout or other factors. At the end of each time step in the markov chain model, there is a possibility that an acquired resistance, $A_i$, may be lost with probability $P_{\text{loss}}(A_i)$. We then simulated the lifetime model with ARG loss (Fig 7(a)) for 1000 individuals for each level of antibiotic use, using the parameter values given in Table 1. A comparison of the results of the simulation with ARG loss, Fig 7(a), and without ARG loss, Fig 2(b), shows negligible differences between the models for each of the three antibiotic use scenarios considered: for low antibiotic use these were mean 7.21 ARG classes with standard deviation 1.95 without gene loss and mean 7.33 standard deviation 1.88 with gene loss; for medium use the means were 9.04 and 9.08 and standard deviations 1.79 and 1.71 respectively; and for high use the means were both 12.63 with

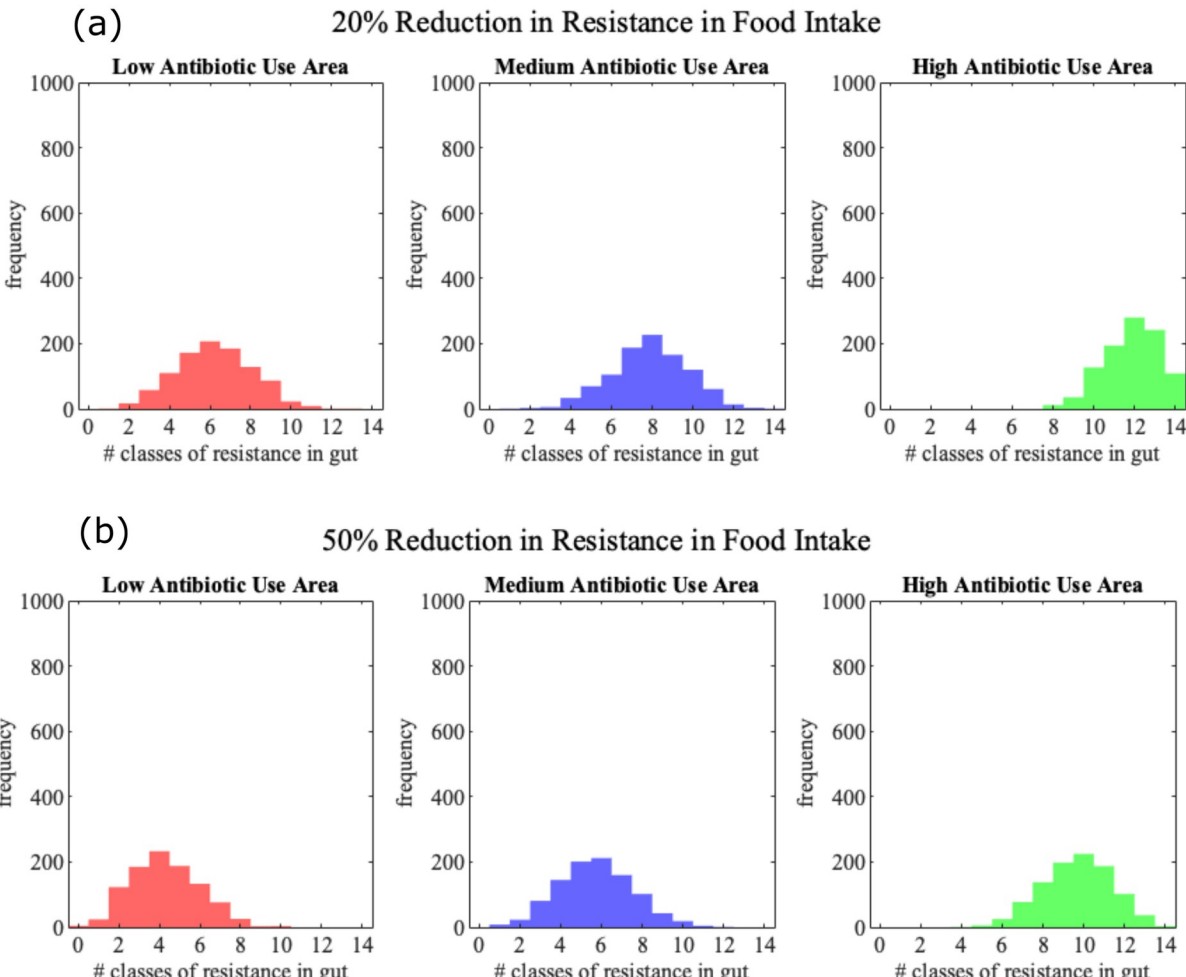

**Fig 5. Histograms showing distribution of ARG load in individual by age 70 under a 20% and 50% reduction in resistance genes in an individual's food intake.** Simulated using 1000 individuals in each of the case for each of the 3 Ab usage areas (low, medium and high). [Mean, Std, Max] for 20% food resistance reduction are [6.2150, 1.8842, 13], [7.8720, 1.9056, 14] and [11.8560, 1.7087, 14] in Low, Medium and High areas respectively. For 50% reduction [4.3020, 1.7099, 10], [5.7340, 1.8522, 12] and [9.677, 1.7087, 14].

standard deviations 1.13 and 1.10 respectively. For all practical purposes these distributions could be considered the same.

We then performed a local sensitivity analysis of the lifetime model with ARG loss to the parameter $P_{\text{loss}}$ (Fig 7(b)). Sensitivity analysis showed that the average resistance load was consistent with the standard lifetime model without ARG loss when the probability of resistance loss was less than $10^{-4}$, while for $P_{\text{loss}}$ greater than $10^{-4}$, we see that the chance of ARG loss is high enough that it leads to a material reduction on the average resistance load. However, it is important to note that this threshold probability of $10^{-4}$ is much higher than we would expect to see for the probability of ARG washout and the physically relevant parameter space for $P_{\text{loss}}$ is expected to be $[10^{-7}, 10^{-5}]$.

## Conclusion

We have demonstrated that three implementable factors can reduce the long-term acquisition and retainment of genes providing resistance to different classes of antibiotics. First,

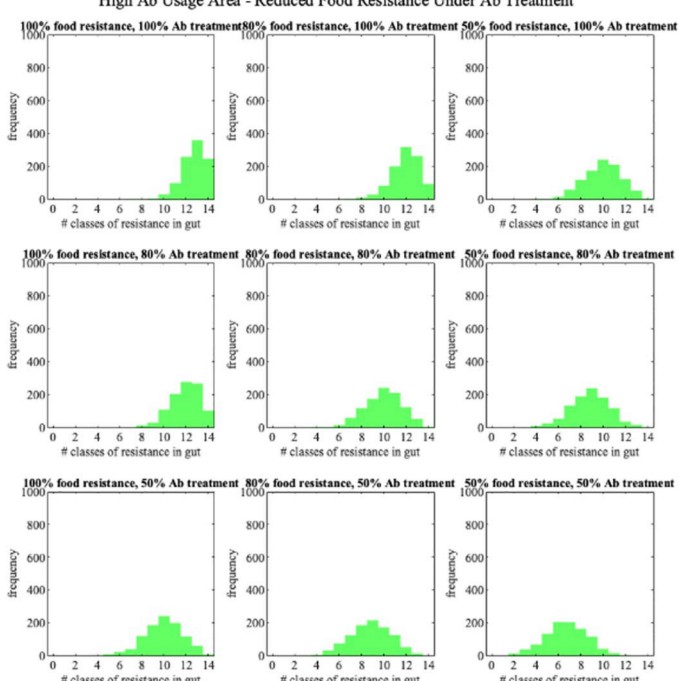

**Fig 6. Distributions of ARG load in resistome for different levels of intervention strategies in high antibiotic usage area.** Simulated using 1000 individuals in each intervention case.

the number of resistance genes acquired by and individual is dependent upon the use of antibiotics over an individual's lifetime. A conservative approach to antibiotic availability and dosing guidelines, as already implemented in many countries, and as advocated in much of literature on antibiotic resistance, would be a practical approach to reducing the long-term number of acquired resistances. Indeed, unnecessary antibiotic treatment can lead to long-term harm to the patient being treated, and could be considered unethical. This argument differs from the more commonly held ethical standpoint that the risk of over-use is primarily to patients other than the one receiving treatment. Second, the number of acquired genes can be reduced even further if an individual's intake of resistance genes, carried on both pathogenic and non-pathogenic bacteria, is also reduced. This could be achieved by policy and practice changes in the food supply chain, including agriculture and post-harvest food production. Third, the reduction in intake of resistance genes is particularly effective during periods of antibiotic treatment where selective pressures increase the likelihood of the retainment of genes. We would suggest that dietary advice should be given to those undergoing antibiotic treatment to avoid products at higher risk of carrying ARGs (even on non-pathogens), as well as ensuring that all food consumed during treatment is fully cooked. The level of benefit to be gained from alterations in medical treatment and dietary changes is highly dependent upon the level of antibiotic use, which varies greatly between countries. While our general model demonstrates benefit across all levels of prescribing, a more nuanced approach that considers region- and country-specific practices, along with specific details of antibiotic classes and associated resistance genes, would provide a better means of quantifying the potential advantages of these changes.

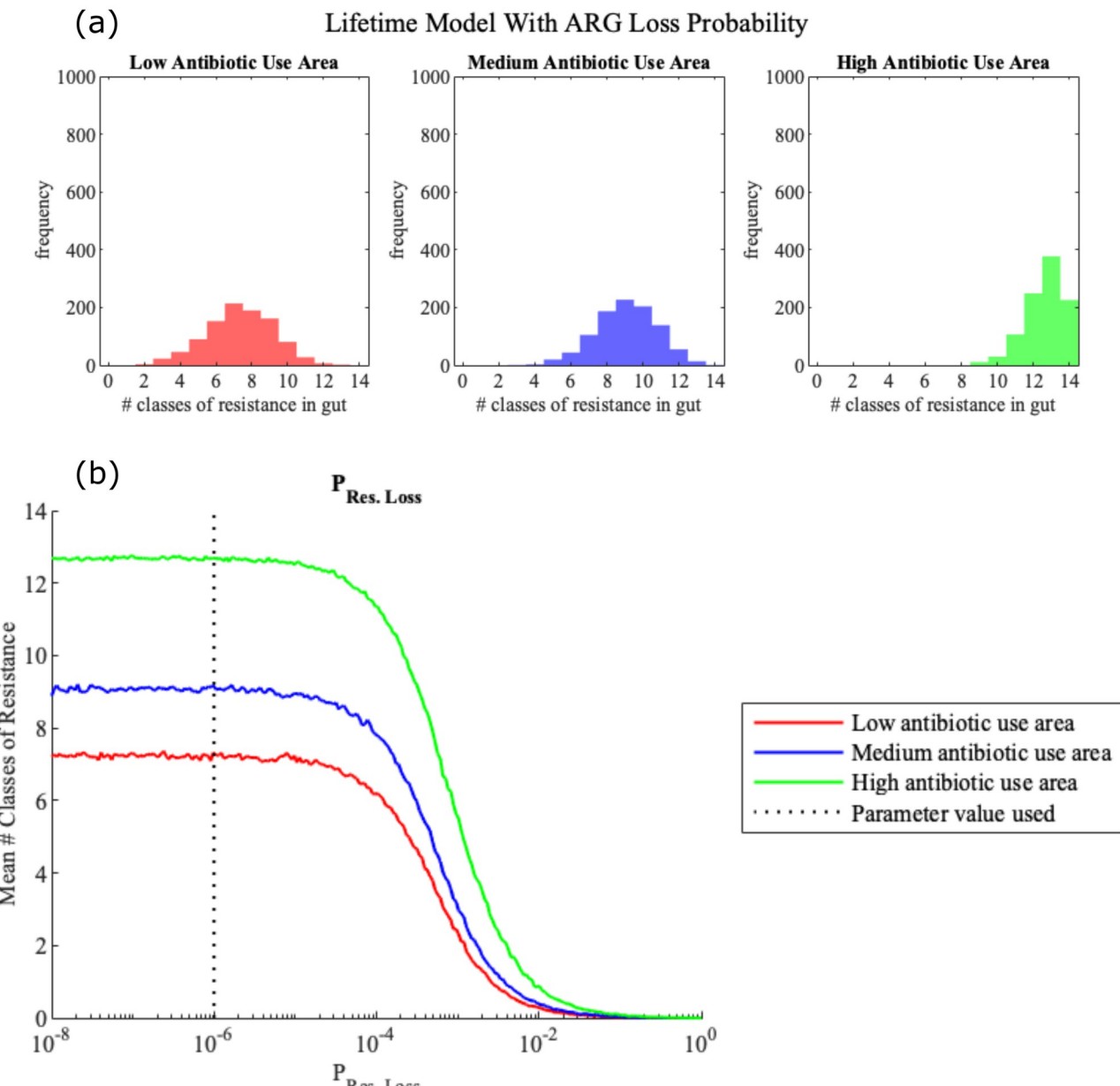

**Fig 7. (a) Histogram showing the distribution of ARG load in individual's resistomes by age 70 for the lifetime resistance model including ARG washout.** These histograms have been made by simulating the lifetime resistance model for 1000 individuals in each of the three antibiotic usage areas, where at each time step in the model there is a possibility that an individual may lose a resistance with probability $P_{Loss} = 1 \times 10^{-6}$. The mean and standard deviation, $(\mu, \sigma)$, for the low, medium and high antibiotic use areas are (7.3330, 1.8841), (9.0780, 1.7131) and (12.6290, 1.1012) respectively. **(b) Local parameter sensitivity analysis of lifetime resistance model to the probability of resistance gene loss**. We vary the value of the probability of an individual losing an acquired resistance, $P_{Loss}$, across the realistic parameter space (given in Table 1) and then calculated the mean ARG load at age 70 of 1000 individuals for each of the different parameter values.

## Supporting information

**S1 Fig. Distributions of ARG load in resistome for different intervention strategies in low antibiotic usage area.**
(TIF)

**S2 Fig. Distributions of ARG load in resistome for different intervention strategies in medium antibiotic usage area.**
(TIF)

**S1 Data. Matlab code for the model used for simulations.**
(M)

## Author Contributions

**Conceptualization:** Dov Joseph Stekel.

**Formal analysis:** Henry Todman, Sankalp Arya.

**Investigation:** Henry Todman, Sankalp Arya.

**Methodology:** Henry Todman, Michelle Baker, Dov Joseph Stekel.

**Supervision:** Michelle Baker, Dov Joseph Stekel.

**Writing – original draft:** Henry Todman.

**Writing – review & editing:** Dov Joseph Stekel.

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
