## [Decision Letter · Decision Letter 0]

28 Jun 2023

PONE-D-23-13150A model of antibiotic resistance genes accumulation through lifetime exposure from food intake and antibiotic treatmentPLOS ONE

Dear Dr. Stekel,

Thank you for submitting your manuscript to PLOS ONE. After careful consideration, we feel that it has merit but does not fully meet PLOS ONE’s publication criteria as it currently stands. Therefore, we invite you to submit a revised version of the manuscript that addresses the points raised during the review process.

Please see below the comments and suggested MINOR revisions made by the individual(s) who reviewed your manuscript.  If provided, the referee's report(s) indicate the revisions that need to be made before it can be accepted for publication.

We look forward to receiving your revised manuscript.

Kind regards,

Ricardo Santos

Academic Editor

PLOS ONE

Reviewers' comments:

Reviewer's Responses to Questions

**Comments to the Author**

1. Is the manuscript technically sound, and do the data support the conclusions?

Reviewer #1: Yes

Reviewer #2: Yes

2. Has the statistical analysis been performed appropriately and rigorously? 

Reviewer #1: I Don't Know

Reviewer #2: N/A

3. Have the authors made all data underlying the findings in their manuscript fully available?

Reviewer #1: Yes

Reviewer #2: Yes

4. Is the manuscript presented in an intelligible fashion and written in standard English?

Reviewer #1: Yes

Reviewer #2: Yes

5. Review Comments to the Author

Reviewer #1: This is an interesting manuscript attempts to model resistance gene accumulation after lifetime exposure to antibiotics through food and treatment with antibiotics. This study sheds new light on AMR. My comments are minor in nature. Typographical errors are outlined on the appended manuscript. See appended manuscript.

Reviewer #2: The manuscript describes the development of a model of ARGs accumulation related to food and antibiotic treatment. ARGs and ARBs spread, and antimicrobial resistances in general, is a very important topic that merits extensive research and the manuscript in itself takes on a very interesting approach. However, there are a few questions that must be addressed before publication:

- the manuscript, although generally well written, as several typos and therefore the authors should make a thorough revision of the manuscript;

- The authors have results and conclusions in the Introduction (page 2, lines 43 - 46). Please remove;

- In the materials and methods, E. coli is used without mentioning it in full before;

- Why the use of 20% and 50% reductions in the different scenarios? These values should be justified

- Figure 4 is presented after line 128, but the explanation comes further down the text, after explaining that there may be other reduction routes, that are for instance shown in Figure 5. The authors should rewrite lines 129-139 so that it is consistent with the presentation of the figures;

- On lines 169-170 and 174-176 where the authors describe negligible differences and significant reduction, respectively, was statistical analysis performed? If so, show the results. If not, statistical analysis should be conducted to improve the manuscript and give stronger meaning to these results.

6. PLOS authors have the option to publish the peer review history of their article (what does this mean?). If published, this will include your full peer review and any attached files.

Reviewer #1: **Yes: **Steven Djordjevic

Reviewer #2: No

---

## [Author Response · Author response to Decision Letter 0]

12 Jul 2023

PONE-D-23-13150

Todman et al. Response to Reviewers

Reviewer #1: This is an interesting manuscript attempts to model resistance gene accumulation after lifetime exposure to antibiotics through food and treatment with antibiotics. This study sheds new light on AMR. My comments are minor in nature. Typographical errors are outlined on the appended manuscript. See appended manuscript.

We thank the reviewer. We have corrected the typographical errors; these are highlighted in the Conclusions sections but not highlighted in references 30, 31 and 33. 

Reviewer #2: The manuscript describes the development of a model of ARGs accumulation related to food and antibiotic treatment. ARGs and ARBs spread, and antimicrobial resistances in general, is a very important topic that merits extensive research and the manuscript in itself takes on a very interesting approach. However, there are a few questions that must be addressed before publication:

- the manuscript, although generally well written, as several typos and therefore the authors should make a thorough revision of the manuscript;

We have checked the manuscript for typographical errors and corrected these, including those noted by Reviewer 1. These have all been highlighted in the manuscript

- The authors have results and conclusions in the Introduction (page 2, lines 43 - 46). Please remove;

We have removed that text and replaced with the following (which is also highlighted in the revised manuscript):

“We use the model to identify which of these factors, individually and in combination, are most important in explaining the accumulation of ARGs during an individual's lifetime, and so identify potential mitigations against this aspect of antimicrobial resistance.”

- In the materials and methods, E. coli is used without mentioning it in full before;

We are not clear whether this referee wants us to spell out Escherichia coli in full or add a line justifying the use of E. coli data. We have done both, as highlighted in the text. The added line is:

“E. coli data are used here because this is a standard sentinel organism used for food safety testing; however, our resistance model is intended to me more generic.”

- Why the use of 20% and 50% reductions in the different scenarios? These values should be justified

This is an interesting point. In the paper we have sought to balance analysis of wide ranges of parameter values (i.e. in the sensitivity analyses) with specific scenarios for which we can draw histograms. 20% and 50% are chosen to represent modest and large reductions at a scale where we would expect to see effect (where say 1% is unlikely). We have made this point clear in the final paragraph of the Methods (highlighted).

- Figure 4 is presented after line 128, but the explanation comes further down the text, after explaining that there may be other reduction routes, that are for instance shown in Figure 5. The authors should rewrite lines 129-139 so that it is consistent with the presentation of the figures;

We thank the reviewer. We have moved text around to ensure a more logical flow. It is highlighted in the text.

- On lines 169-170 and 174-176 where the authors describe negligible differences and significant reduction, respectively, was statistical analysis performed? If so, show the results. If not, statistical analysis should be conducted to improve the manuscript and give stronger meaning to these results.

We thank the reviewer for this point. The language we used here was a imprecise. We have changed the word ‘significantly’ with ‘materially’ in the sub-heading and in the description of the sensitivity analysis. With regards the first comparison, we have added text to describe the summary statistics of the distributions obtained which makes clear that the differences between the distributions are negligible.

---

## [Decision Letter · Decision Letter 1]

31 Jul 2023

A model of antibiotic resistance genes accumulation through lifetime exposure from food intake and antibiotic treatment

PONE-D-23-13150R1

Dear Dr. Stekel,

We’re pleased to inform you that your manuscript has been judged scientifically suitable for publication and will be formally accepted for publication once it meets all outstanding technical requirements.

Kind regards,

Ricardo Santos

Academic Editor

PLOS ONE

Additional Editor Comments (optional):

Reviewers' comments:

Reviewer's Responses to Questions

**Comments to the Author**

1. If the authors have adequately addressed your comments raised in a previous round of review and you feel that this manuscript is now acceptable for publication, you may indicate that here to bypass the “Comments to the Author” section, enter your conflict of interest statement in the “Confidential to Editor” section, and submit your "Accept" recommendation.

Reviewer #1: All comments have been addressed

2. Is the manuscript technically sound, and do the data support the conclusions?

Reviewer #1: Yes

3. Has the statistical analysis been performed appropriately and rigorously? 

Reviewer #1: Yes

4. Have the authors made all data underlying the findings in their manuscript fully available?

Reviewer #1: Yes

5. Is the manuscript presented in an intelligible fashion and written in standard English?

Reviewer #1: Yes

6. Review Comments to the Author

Reviewer #1: I am satisfied that the authors have made the appropriate corrections in the revised manuscript. I have no further queries and recommend that the manuscript be accepted for publication.

7. PLOS authors have the option to publish the peer review history of their article (what does this mean?). If published, this will include your full peer review and any attached files.

Reviewer #1: **Yes: **Steven P. Djordjevic

---

## [Editor Report · Acceptance letter]

8 Aug 2023

PONE-D-23-13150R1 

A model of antibiotic resistance genes accumulation through lifetime exposure from food intake and antibiotic treatment 

Dear Dr. Stekel:

I'm pleased to inform you that your manuscript has been deemed suitable for publication in PLOS ONE. Congratulations! Your manuscript is now with our production department. 

Kind regards, 

on behalf of

Dr. Ricardo Santos 

Academic Editor

PLOS ONE